# Anthropogenic Disturbances Influenced the Island Effect on Both Taxonomic and Phylogenetic Diversity on Subtropical Islands, Pingtan, China

**DOI:** 10.3390/plants13111537

**Published:** 2024-06-01

**Authors:** Bo Chen, Denghong Xue, Zhihui Li, Lan Jiang, Yu Tian, Jing Zhu, Xing Jin, Jingjing Yang, Chaofa Huang, Jurong Liu, Hai Liu, Jinfu Liu, Zhongsheng He

**Affiliations:** 1Key Laboratory of Fujian Universities for Ecology and Resource Statistics, College of Forestry, Fujian Agriculture and Forestry University, Fuzhou 350002, China; cb137751120@126.com (B.C.); lzhihui9801@163.com (Z.L.); jlnaruto0629@126.com (L.J.); hy290527304@126.com (Y.T.); hdly0718@126.com (J.Z.); fjnldxjinxing@163.com (X.J.); yangjingjingfafu@163.com (J.Y.); 2Pingtan Comprehensive Experimental Zone Natural Resources Service Center, Fuzhou 350400, China; ptlyjzyz@126.com; 3Fujian Forestry Prospect and Design Institute, Fuzhou 350001, China; 15425811@126.com; 4Fujian Forestry Survey and Planning Institute, Fuzhou 350001, China; 49847396@126.com (J.L.); hl272949341@126.com (H.L.)

**Keywords:** taxonomic diversity, phylogenetic diversity, island effects, habitat heterogeneity, anthropogenic disturbances

## Abstract

The investigation of taxonomic diversity within island plant communities stands as a central focus in the field of island biogeography. Phylogenetic diversity is crucial for unraveling the evolutionary history, ecological functions, and species combinations within island plant communities. Island effects (area and isolation effect) may shape species distribution patterns, habitat heterogeneity affects habitat diversity, and anthropogenic disturbances can lead to species extinction and habitat destruction, thus impacting both species diversity and phylogenetic diversity. To investigate how taxonomic and phylogenetic diversity in island natural plant communities respond to island effects, habitat heterogeneity, and anthropogenic disturbances, we took the main island of Haitan (a land-bridge island) and nine surrounding islands (oceanic islands) of varying sizes as the subjects of our study on the Pingtan islands. We aim to elucidate the influence of island effects, habitat heterogeneity, and anthropogenic disturbances on taxonomic and phylogenetic diversity. The results showed that, (1) Both the taxonomic and phylogenetic diversity of plants on the Pingtan islands followed the island area effect, indicating that as the island area increases, both taxonomic and phylogenetic diversity also increase. (2) Island effects and habitat heterogeneity were found to enhance taxonomic and phylogenetic diversity, whereas anthropogenic disturbances were associated with a decrease in both taxonomic and phylogenetic diversity. Furthermore, the synergistic influence of island effects, habitat heterogeneity, and anthropogenic disturbances collectively exerted a negative impact on both taxonomic and phylogenetic diversity. (3) The contribution of explanatory variables of anthropogenic disturbances for taxonomic and phylogenetic diversity was higher than that of island effects and habitat heterogeneity. Additionally, the contribution of the explanatory variables under the combined influence of island effects, habitat heterogeneity, and anthropogenic disturbances is higher than that of the individual variables for island effects and habitat heterogeneity. These findings suggest that anthropogenic disturbances emerged as the dominant factors influencing both taxonomic and phylogenetic diversity. These findings demonstrate the intricate interplay between island effects, habitat heterogeneity, and anthropogenic disturbances, highlighting their combined influence on both taxonomic and phylogenetic diversity on island.

## 1. Introduction

Island ecosystems are characterized by resource uniqueness, system integrity, and ecological vulnerability [1]. Geographic isolation limits the exchange and circulation of materials and energy with external environments, fostering the development of unique habitats and vegetation communities [2]. The early Equilibrium Theory of Island Biogeography (ETIB) provides a theoretical framework for exploring biodiversity in island ecosystems, emphasizing the critical role of island size and proximity to the mainland in determining species richness [3]. Resource distribution within islands correlates with island size; larger islands possess more ecological niches and resources, supporting a higher diversity of species. Additionally, isolation reduces colonization rates, and greater isolation enhances the likelihood of species extinction [4]. Island spatial characteristics include distinct geographic, topographical, and ecological attributes that result in unique ecosystem structures [5]. The interplay between these characteristics and anthropogenic disturbances is significant; the unique geographic and ecological traits of islands increase their vulnerability to human impacts, which in turn shape island ecosystem patterns [6]. These unique geographical features make each island a distinctive sample for biodiversity research. Although the ETIB provides a basic framework for understanding the relationship between species richness and island size, as well as distance from the mainland, the theory has its limitations. It fails to fully account for the impacts of characteristic factors such as habitat heterogeneity on islands and human disturbances on diversity [3]. Therefore, integrating ETIB into a comprehensive study of island biodiversity, particularly emphasizing the island effect and fully considering characteristic factors such as island habitat heterogeneity and anthropogenic disturbances, contributes to an understanding of the maintenance of island biodiversity [2,7].

The primary goal of biodiversity conservation is not merely to increase species numbers but also to preserve the evolutionary information inherent in various species [8]. While the ETIB primarily focuses on species richness, phylogenetic diversity considers the evolutionary history, ecological functions, and species combinations within island plant communities [9,10,11]. From an evolutionary perspective, the importance of biodiversity conservation varies among species within communities, where the current pattern of biodiversity distribution is an outcome of a series of evolutionary processes [12]. In scenarios with limited resources, priority should be accorded to protecting groups that are evolutionarily distinct from other species [12]. Community phylogeny, which includes phylogenetic relationships and evolutionary data among species, forms a crucial aspect of biodiversity [13]. It addresses the limitations of taxonomic diversity measures that do not capture the evolutionary details of community species [13,14]. Studies such as those of Forest et al. and Pio et al. at Cape Good Hope demonstrate that phylogenetic diversity is a more effective reference criterion than species richness for designing plant diversity conservation strategies [15,16]. Further investigations reveal that phylogenetic relationships among habitat specialists on terrestrial islands have minor impacts on specialized plant communities, increasing clustering with isolation [17]. Historical dynamics of island topography influence phylogenetic and taxonomic diversity through habitat heterogeneity, with larger islands exhibiting higher phylogenetic diversity [18,19]. While many studies often focus on investigating the impact of individual island characteristics (such as size, shape, and geographic location) on phylogenetic diversity, comprehensive research considering multiple island features is crucial for a more holistic understanding of ecosystem complexity and species formation and maintenance [5,20,21]. Additionally, human activities have significantly altered the formation of many island species [22,23,24]. Island community phylogenetics was also shaped by habitat heterogeneity, with human activities potentially altering species evolution [6]. Therefore, comprehensive research considering multiple island features and human activities is crucial for understanding the shaping of island phylogenetic diversity.

Currently, most ETIB studies focus primarily on validating species richness, which only considers the quantity of species and may not comprehensively reflect the complexity of ecosystems, but as indicated by the Shannon–Wiener, Simpson, Margalef, and Pielou indices, phylogenetic diversity and phylogenetic structure might be more representative. For example, the Shannon–Wiener index considers both species richness and evenness, providing a more comprehensive assessment of diversity. Similarly, the Simpson index focuses on dominance within a community, while the Margalef index accounts for species richness relative to the total number of individuals [5,8]. Both community taxonomic and phylogenetic diversity are not exclusively influenced by island effects, geographic isolation, and habitat heterogeneity, and human activities may also affect the island taxonomic and phylogenetic diversity. Based on the above description, our study addressed two main questions, (1) What pattern of taxonomic and phylogenetic diversity is observed with island area and isolation? (2) How do the island effect, island habitat heterogeneity, anthropogenic disturbances, and the combination of these factors influence taxonomic and phylogenetic diversity? We hypothesize that taxonomic and phylogenetic diversity will increase with island area and decrease with isolation, and that the island effect, island habitat heterogeneity, anthropogenic disturbances and the combination of these factors can influence taxonomic and phylogenetic diversity, while anthropogenic disturbances could affect the island characteristics on both taxonomic and phylogenetic diversity.

## 2. Result

### 2.1. The Relationship between Island Area and Taxonomic, Phylogenetic Diversity, and Structure

The island area was significantly positively correlated with species richness (SR), as well as the Shannon–Wiener, Pielou, Simpson, Margalef, and phylogenetic (*SES.PD*) diversity indices (*p* < 0.05) (Figure 1). As the island area increases, these indices also increase, with no significant correlation observed with *SES.MPD* and *SES.MNTD* indices (Appendix A). Island isolation showed no significant correlation with taxonomic, phylogenetic diversity indices, and phylogenetic structure. However, as isolation increases, there was a decreasing trend in both taxonomic and phylogenetic diversity indices (Appendix A).

### 2.2. Relationships between Island Factors and Island Taxonomic, Phylogenetic Diversity, and Structure

The results of the first principal component (PCA) scores for different combinations, including G1, G2, G3, and their combinations, can be found in Appendix A. G1 showed a significant positive effect (*p* < 0.01) on SR, *SES.PD*, Shannon–Wiener, Pielou, and Margalef indices. G3 showed a significant negative effect (*p* < 0.01) on SR, *SES.PD*, Shannon–Wiener, Pielou, and Margalef indices. G12 and G13 showed significant negative effects (*p* < 0.05) on *SES.MPD*, Shannon–Wiener, and Pielou indices. G23 showed a significant negative effect on SR, *SES.PD*, and Margalef indices. G123 showed a significant negative effect (*p* < 0.001) on SR, *SES.PD*, and Margalef indices (Table 1).

### 2.3. The Driving Factors of Taxonomic, Phylogenetic Diversity and Structure

The contribution of explanatory variables of different factor combinations on taxonomic and phylogenetic diversity is presented in Figure 2. G1, G3, G23, and G123 contributed explanatory percentages of 17.75%, 27.88%, 23.31%, and 31.05% to SR, respectively (Figure 2a). The explanatory percentages of G1, G3, G23, and G123 on *SES.PD* were 20.35%, 27.76%, 26.06%, and 25.83%, respectively (Figure 2b). For Margalef, the explanatory percentages were 21.20%, 26.83%, 26.25%, and 25.72% for G1, G3, G23, and G123, respectively (Figure 2f). The explanatory percentages of G1, G2, G3, G12, and G13 on the Shannon–Wiener index were 18.64%, 17.63%, 21.08%, 17.23%, and 25.42%, respectively (Figure 2c). Similarly, for the Pielou index, the explanatory percentages were 18.33%, 16.63%, 20.07%, 18.65%, and 26.32%, respectively (Figure 2d). The explanatory percentages for the Simpson index were 31.73%, 33.08%, and 35.19% for G2, G12, and G13, respectively (Figure 2e). Regarding the *SES.MPD* index, the explanatory percentages are 51.84% and 48.16% for G12 and G13, respectively (Figure 2g).

## 3. Discussion

### 3.1. The Impact of Island Area and Isolation on Taxonomic, Phylogenetic Diversity, and Structure

On the Pingtan islands, species richness (SR) showed a significant positive correlation with *SES.PD*, Shannon–Wiener, Pielou, and Margalef indices (*p* < 0.05) (Appendix A). Moreover, SR and Shannon–Wiener, Pielou, Simpson, and Margalef indices showed a significant positive correlation with island area (*p* < 0.05) (Figure 1). As island area increases, species richness also increases, with larger islands having higher species richness. This study’s results align with the first hypothesis, specifically the presence of the island area effect, indicating a substantial association between richness and other measures of taxonomic diversity. Besides SR, Shannon–Wiener, Pielou, and *SES.PD* indices also demonstrate the island area effect. This finding was consistent with the well-established island area effect widely studied in island biogeography [3]. The study results indicate a gradual increase in species richness with the enlargement of island area. This phenomenon can be explained by the notion that larger islands may offer a more diverse range of habitats and resources, providing ample opportunities for various plant species to settle and coexist [25,26].

On the Pingtan islands, there was no clear linear relationship between island taxonomic and phylogenetic diversity, and phylogenetic structure with isolation. However, there was a tendency for a decrease in taxonomic and phylogenetic diversity, and phylogenetic structure (Appendix A). The differences between the results of this study and some previous research could be attributed to several factors. On one hand, the lack of a clear relationship between isolation and taxonomic and phylogenetic diversity, and phylogenetic structure may be due to the relatively similar or unclear isolation levels among different islands on the Pingtan islands (Appendix A). On the other hand, a more significant factor could be the influence of anthropogenic disturbances, which might disrupt the expected isolation effect among islands [6,10,27].

### 3.2. The Analysis of the Impact of Island Effects, Island Habitat Heterogeneity, and Anthropogenic Disturbances on Taxonomic, Phylogenetic Diversity, and Structure

Islands represent distinctive geographical entities marked by discernible island effects and habitat heterogeneity. The ongoing escalation of human activities perpetually disrupts the stability of taxonomic diversity and exerts an influence on species evolution [28]. Generally, the larger the island area, the higher the species diversity, the greater the isolation, and the lower the species diversity [29]. The island effect (PCA-G1) has positive significance between SR, *SES.PD*, Shannon–Wiener, Pielou, and Margalef indices on the Pingtan islands, indicating that island area has a greater impact on species diversity compared to island isolation. Large and relatively close islands often support richer biodiversity, and our previous research on the area effect exhibited by the Pingtan islands also indicated this. This suggests that within the island environment, the complex interplay of factors such as habitat diversity, resource availability, migration connectivity, and interspecies combinations promotes adaptation and proliferation of species on the islands. As a result, there was a significant increase in SR, *SES.PD*, evenness, and overall ecosystem richness [30].

Positive significance has also been found between island habitat heterogeneity (PCA-G2) and Shannon–Wiener, Pielou, and Simpson indices. This result was consistent with a study by Li et al. [20] on the spatial patterns of plant distribution and island shape effects in the Zhoushan Archipelago, indicating a close correlation between the complexity of island shapes and ecosystem diversity and structure. On one hand, the intricate shapes of islands may provide richer ecological niches and micro-environments, creating various habitat types. On the other hand, complex shapes may result in longer edges and more migration pathways, reinforcing combinations and migration between species, contributing to maintaining high levels of richness [2,31].

Negative significance has been found between anthropogenic disturbances (PCA-G3) and SR, *SES.PD*, Shannon–Wiener, Pielou, and Margalef indices. This was consistent with the findings of many studies on the impact of human activities on island taxonomic diversity [32,33]. The possible reasons stem from the fact that human disturbance can directly or indirectly result in habitat destruction, the introduction of invasive species, resource competition, habitat fragmentation, and disruption of ecosystem stability [34]. These factors, in turn, impact biodiversity and species evolution in various ways [34].

We combined island characteristics, habitat differences, and human impacts. We found that G23 had a negative effect on species richness (SR), phylogenetic diversity (*SES.PD*), and the Margalef index. This shows the combined influence of island features, which create diverse habitats, and human activities, which bring external disturbances and alter habitats. Collectively, these factors result in negative responses in the ecosystem, including species loss and a reduction in phylogenetic diversity [35]. G123 showed a significant negative impact on SR, *SES.PD*, and the Margalef index. This indicates that island ecosystems are influenced by multiple factors, including heterogeneity in internal island structure, island effects, and anthropogenic disturbances, leading to overall ecosystem instability and a decrease in diversity [10]. These results aligned with our second hypothesis, suggesting that not only individual factors such as island effects, island habitat heterogeneity, and anthropogenic disturbances independently impact diversity, but their combinations also influence community structure and species evolution. The study results further indicate that, mediated by human activities, the combined impact of island effects, island habitat heterogeneity, and anthropogenic disturbances show a significant negative effect on relevant ecological indicators (Table 1). This reflects that, under the influence of human activities, the joint action of these factors leads to a negative response in the ecosystem [28].

### 3.3. The Relative Importance Analysis of Island Effects, Island Habitat Heterogeneity, and Anthropogenic Disturbances on Taxonomic, Phylogenetic Diversity, and Structure

Islands typically display unique island effects and spatial characteristics, which influence the distribution and evolutionary relationships of species. Although human activities further impact the distribution and evolution of species, the trade-off relationship among these three factors in influencing taxonomic diversity is inconsistent [36,37]. In this study, the contribution of explanatory variables for G1, G3, G23, and G123 on SR, *SES.PD*, and Margalef decreases in the following order, G1 (positive effect) < G3 (negative effect) < G23 (negative effect) < G123 (negative effect) (Figure 2a,b,f). The contribution of explanatory variables of G1 (positive effect) was the smallest. This result was similar to the study by Wohlwend et al. [38] on the impact of human activities on plant diversity on different islands in the Pacific, which showed that human interference has a greater impact on plant phylogeny compared to island habitat heterogeneity. This was because large islands often have a relatively unique ecological environment, which may provide more opportunities and resources for species, promoting species richness [39]. However, island effects may be constrained by other factors, such as human activities often accompanying habitat destruction, pollution, and climate change. These factors have a negative impact on the survival and reproduction of species, leading to a higher degree of negative impact on species richness than island effects [40]. The negative effect contribution of explanatory variables of G123 was relatively high. This is because the positive effects of island effects and habitat heterogeneity on diversity are gradually constrained in the face of human activities, with anthropogenic disturbances becoming a more dominant factor, leading to the corresponding change in the contribution of explanatory variables trend [5]. In comparison to previous studies that primarily focused on the impact of individual island effects or anthropogenic disturbances factors on island diversity, our research comprehensively considers the complex combinations of these factors [10,41]. In addition, our study revealed that when combining individual island effects, habitat heterogeneity, and anthropogenic disturbances, they collectively exerted negative effects, with anthropogenic disturbances playing a predominant role (Figure 2a–f). This was attributed to factors such as the proximity to Haitan Island and the extent of land use, which were strongly associated with human activities and habitat degradation. Haitan Island, being a hub for human activities, significantly impacts plant diversity in the surrounding islands, particularly those closer to it, which are more susceptible to human-induced disturbances [42]. These findings support our hypothesis that anthropogenic disturbances diminish the influence of island characteristics on taxonomic and phylogenetic diversity.

Significant limitations arise from data collected from only nine islands, given the current absence of additional data. It is imperative to acknowledge that this limitation may impact the accuracy and reliability of this study’s conclusions. The small sample size may heighten the instability and randomness of the results, thus diminishing the study’s reliability. Additionally, due to the restricted dataset, we may not have fully considered all potential variations and influencing factors, potentially resulting in imprecise conclusions or deviations from reality. But, through this research, we have been able to reveal some key patterns and trends in the biodiversity of island ecosystems. We have found that island effects, habitat heterogeneity, and anthropogenic disturbances play crucial roles in shaping plant diversity. Despite the limited data sources, we have endeavored to provide detailed explanations of these findings in the discussion, highlighting possible interpretations and directions for future research. Therefore, we plan to supplement more island sampling sites in subsequent research to increase the sample size and enhance the credibility and representativeness of this study.

## 4. Materials and Methods

### 4.1. Study Site

Pingtan island is located in the eastern sea area of Fujian Province, China (119°32′26″~120°11′56″ E, 25°15′43″~25°40′10″ N) (Figure 3). It is situated in the subtropical maritime monsoon climate zone. The island is approximately 29 km long from north to south and 19 km wide from east to west. The average annual temperature is around 19.6 °C. The average temperatures on the hottest and coldest days are approximately 27.9 °C and 10.2 °C, respectively. The annual average rainfall is 1172 mm, slightly lower than the annual average evaporation (1300 mm). Prolonged high temperatures and drought conditions are more common during the period from July to September. The summer is mainly influenced by the south wind, while the autumn, winter, and spring seasons are predominantly influenced by the northeast wind. The annual average wind speed is approximately 6.9 m/s, with more than 125 days of windy weather (7 or above on the Beaufort scale), often affected by typhoon conditions. The soils on Haitan Island are primarily composed of brick-red ferrosol, coastal sandy soil, and saline soil, with relatively low soil fertility [43]. Around Haitan Island, there are several other islands such as Dalian Island, Xiaolian Island, Huangmen Island, Monkey Island, and Beiguan Island, with a total forest area of 12,367 hm^2^, total vegetation area of 11,902 hm^2^, and total vegetation coverage rate of 30.06% on the Pingtan islands. The primary tree species of the Pingtan islands include *Pinus elliottii*, *P. massoniana*, *Casuarina equisetifolia*, and *Cyclobalanopsis chungii*, among others [44,45].

### 4.2. Sample Plot Setting

Haitan Island is the main island in Pingtan. In April 2023, typical plots were delineated on Haitan Island and nine surrounding islands. The selection of these plots was based on the vegetation coverage of the islands [46]. We initially utilized remote sensing imagery to assess the density and coverage of vegetation on the islands. Based on this evaluation and considering the island areas, we established multiple sampling sites across the 9 islands. We implemented a gradient approach, categorizing the islands into large, medium, and small based on their area. Accordingly, we set up 9, 6, and 3 sampling sites for large, medium, and small islands, respectively. However, we observed that larger islands did not necessarily exhibit the highest vegetation coverage. Therefore, we adjusted our sampling strategy to proportionally match the number of sampling sites with the vegetation coverage, aiming to ensure that each island’s sampling sites are representative of its size and habitat diversity as much as possible.

We established six 10 m × 10 m quadrats at each of the three locations on Haitan Island, Da mu (119°44′ E, 25°27′ N), Xianjian Country (119°42′ E, 25°29′ N), and Junshan Mountain (119°48′ E, 25°35′ N). At the surrounding islands, namely Dalian Island (119°40′ E, 25°38′ N), Cao Island (119°43′ E, 25°23′ N), and Monkey Island (119°40′ E, 25°29′ N), each location had nine 10 m × 10 m quadrats. At Jiangshan Island (119°48′ E, 25°26′ N), Beiguan Island (119°39′ E, 25°22′ N), and Huangmen Island (119°40′ E, 25°27′ N), each location had six 10 m × 10 m quadrats. Finally, at Nanguan Island (119°40′ E, 25°20′ N), Baijiang Island (119°48′ E, 25°25′ N), and Qing Island (119°48′ E, 25°31′ N), each location had three 10 m × 10 m quadrats (refer to Figure 3 and Appendix A for details). The species within all sample plots are determined to be wild plants based on observations of their growth environment, growth morphology, and historical records from the Natural Resources and Ecology Environment Bureau of Pingtan Comprehensive Experimental Zone in Fujian Province, China, indicating that these plots have not been subjected to deliberate planting, management, or marking by humans. For all trees with a diameter at breast height (DBH) ≥ 1 cm within the sample plots, individual measurements were taken, including the scientific name, DBH, and tree height. Each 10 m × 10 m plot was subdivided into four 5 m × 5 m subplots. Within each 5 m × 5 m subplot, five 1 m × 1 m quadrats were established (east, west, south, north, and center) to investigate the herbaceous layer. Information such as species names, height, and coverage of herbaceous plants was recorded within these 1 m × 1 m quadrats. Plant species names were documented with reference to the online Flora of China (http://frps.iplant.cn/ (accessed on 3 July 2023)).

### 4.3. Taxonomic, Phylogenetic Diversity, and Phylogenetic Structure

The α diversity index encompasses various metrics including species richness (SR), the Shannon–Wiener index, the Simpson index, the Margalef index, and the Pielou index. The Shannon–Wiener index assumes random sampling of individuals within an infinitely large community, where the probability of encountering an individual in the sample reflects the diversity index [47]. The calculation formula is as follows:$$H^{\prime }=-\sum_{i=1}^{s}\left(P_{i}\ln P_{i}\right)$$
where *P_i_* = *N_i_*/*N* is the proportion of individuals of species *i* relative to the total number of individuals and *S* is the total number of species.

Similarly, the Simpson index measures diversity by considering random sampling of individuals within infinite communities, where the probability of encountering two individuals of different species serves as the diversity metric [48]. The calculation formula is as follows:$$D=\sum_{i=1}^{s}{\left(P_{i}\right)}^{2}$$
where *P_i_* = *N_i_*/*N* indicates the ratio of individuals of species *i* to the total number of individuals, and *S* refers to the total count of species.

The Margalef index focuses solely on the count of species and total individuals in a community, defining the species count in a sample of a certain size as the diversity index [49]. The calculation formula is as follows:$$d=\frac{S-1}{lnN}$$
where *N* represents the total number of individuals, *S* represents the total number of species, and ln is the natural logarithm base.

The Pielou index represents an evenness index independent of species richness [49]. The calculation formula is as follows:$$J^{\prime }=\frac{H^{\prime }}{Hmax}\;\;$$
where *H*′ is the Shannon–Wiener index, and *Hmax* = *lnS* is the maximum possible value of *H*′ for the community.

After obtaining the list of taxa at the level of family, genus, and species, we used the R 4.0.5 *plantlist* package for the species evolutionary tree. The plant classification system APGIII, along with lineage tree structure data incorporating differentiation times, was generated online using phylomatic (http://phylodiversity.net/phylomatic/ (accessed on 23 October 2023)). The Zanne evolutionary tree framework was then integrated. Subsequently, the picante package was employed to calculate phylogenetic diversity and structure [50]. Phylogenetic diversity (*PD*) index was utilized for measuring the total branch length of community development. To ensure the randomness of species individuals in their spatial distribution, we utilized the “taxa.labels” null model provided by the “*picante*” package in R 4.0.5 software for our analysis. This null model shuffles species occurrences while preserving the total species richness and species abundances, thereby ensuring the randomness of species individuals in their spatial distribution [51].

*SES.MPD* (Standardized Effect Size of Mean Pairwise Distance) is a standardized metric used to assess the deviation between the mean phylogenetic distance among species in a community and the expected distance under a random model. It reflects whether the average evolutionary distance among species is closer or farther than expected by random selection. A significantly positive *SES.MPD* value indicates that species in the community are closer in phylogeny than randomly selected species, suggesting a strong environmental filtering favoring ecologically similar species. Conversely, a significantly negative *SES.MPD* value indicates that species in the community are more phylogenetically dispersed than randomly selected species, possibly indicating competitive exclusion, where closely related species struggle to coexist due to resource competition. *SES.MNTD* (Standardized Effect Size of Mean Nearest Taxon Distance) is also a standardized metric used to assess the deviation between the mean phylogenetic distance from each species to its nearest taxon and the expected distance under a random model. It reflects whether the average evolutionary distance between species and their nearest taxon is closer or farther than expected by random selection. A significantly positive *SES.MNTD* value indicates that species in the community are closer in phylogeny to their nearest taxon than randomly selected species, suggesting stricter environmental filtering. Conversely, a significantly negative *SES.MNTD* value indicates that species in the community are more phylogenetically dispersed from their nearest taxon than randomly selected species, suggesting a more pronounced competitive exclusion at local scales [52,53]. The calculation formulas for *SES.MPD* and *SES.MNTD* are as follows,
(1)$$SES.MPD=\frac{{MPD}_{obs\;}-{MPD}_{null}}{SD({MPD}_{null})}$$
(2)$$SES.MNTD=\frac{{MNTD}_{obs}-{MNTD}_{null}}{SD({MNTD}_{null})}$$
where *MPD_obs_* is the observed average evolutionary distance between species, *MPD_null_* is the average value of the average evolutionary distance between species under the zero model, and *SD* (*MPD_null_*) is the standard deviation of the average evolutionary distance between species under the zero model. For Equation (2), *MNTD_obs_* is the observed average evolutionary distance between the nearest species, *MNTD_null_* is the average evolutionary distance between the nearest species under the zero model, and *SD* (*MNTD_null_*) is the standard deviation of the average evolutionary distance between the nearest species under the zero model [52].

In order to standardize the *SES.PD*, Margalef, Simpson, Shannon–Wiener, and Pielou indices and species richness (SR) for each island, we employed a method based on species–area accumulation curves. Firstly, we plotted species–area accumulation curves based on the sample plot data for each island. These curves illustrate the number of new species discovered with increasing sample plot area. Subsequently, we calculated the species accumulation rate for each island, representing the number of new species discovered per unit area. Finally, we divided each index value by the logarithm (with a base of 10) of the sample plot area to account for differences in area among islands [39,54]. It is noteworthy that *SES.MPD* and *SES.MNTD*, being standardized indices, were not subjected to this transformation in the study [54]. The taxonomic and phylogenetic diversity indices for different-sized islands on the Pingtan islands are presented in Appendix A.

### 4.4. Degree of Anthropogenic Disturbance, and Island Spatial Characteristics

The characteristics of the surveyed islands were calculated by Geographic Information System (GIS) 3.34.7 software. Parameters such as island area, coastline length, distance to the mainland, nearest island distance, island elevation, and shape index (SI) were used to depict the island features. Specifically, island area was determined by spatial analysis tools within GIS that compute the total area based on the island’ s boundary coordinates. Coastline length was measured using GIS tools that trace the contour lines of the island perimeter, ensuring an accurate reflection of the island’ s interface with water bodies. The distance to the mainland (MD, distance to the mainland) and nearest island distance—measuring the shortest straight-line distance from a particular island to its nearest neighboring island—were calculated using point-to-point measurement functionalities in GIS. Island elevation was assessed by integrating GIS with terrain data, such as Digital Elevation Models (DEM), to extract the highest elevation point on the island. The isolation of the islands was assessed using the distance to the mainland. This metric, along with the nearest island distance, plays a crucial role in assessing island spatial characteristics as it provides information about the proximity of one island to another, which can influence ecological processes such as dispersal and species interactions [55]. The shape index (SI) reflects the complexity of the island’s shape and was calculated using the formula SI = Per/[2 × (π × *A*)^0.5^], where Per is the coastline length of the island, and A is the island area [56]. To calculate the area of different land use types on Pingtan Island, we first obtained land use data for the area and then preprocessed and analyzed the data using Geographic Information System (GIS) software. During the analysis process, the land was classified into different categories, including farmland and developed land. Subsequently, for each category, the total area within the study area was calculated using ArcGIS tools. The detail values refer to Appendix A.

### 4.5. Classification of Island Effect, Habitat Heterogeneity and Anthropogenic Disturbance

Island effects may influence both species diversity and phylogenetic diversity, as the geographical isolation and constraints of islands can shape species distribution patterns and phylogenetic relationships. Habitat heterogeneity can result in different environmental conditions and habitat types across regions, affecting species survival and adaptation, thereby influencing both species diversity and phylogenetic diversity. Anthropogenic disturbances may disrupt the balance of ecosystems, leading to species extinction, habitat destruction, and population fragmentation, thus impacting both species diversity and phylogenetic diversity. To explore the island effect and its impact on diversity, considering the habitat heterogeneity, human interference, and the combinations between these factors, we categorized all influencing factors into three groups. The first group is the island effect (G1, Group 1), encompassing conventional island effects such as island area and isolation effects. Therefore, we classified area and isolation (distance from the mainland) as factors contributing to the island effect. The second group consists of island habitat heterogeneity factors (G2, Group 2), including coastline length, maximum elevation, nearest island distance, and shape index, based on the inherent attributes of island space. The third group is anthropogenic disturbance (G3, Group 3). As Haitan Island experiences the highest frequency of human activities and the highest proportion of land use compared to other islands, proximity to Haitan Island is considered a factor increasing susceptibility to human-induced disturbances. Therefore, we categorized the distance from different islands to Haitan Island and the proportion of land use on other islands as factors influencing human-induced disturbance. The combinations of different island effects, island habitat heterogeneity, and anthropogenic disturbances factors are divided into seven categories: island effect (G1), island habitat heterogeneity (G2), anthropogenic disturbances (G3), island Effect + island habitat heterogeneity (G12), island effect + anthropogenic disturbances (G13), island habitat heterogeneity + anthropogenic disturbances (G23), and island effect + island habitat heterogeneity + anthropogenic disturbances (G123). To ascertain the response variables of various combinations, we conducted PCA (Principal Component Analysis) and observed that the factors of different combinations predominantly aligned with the first axis, which accounted for the highest percentage of explained variance. Consequently, we utilized the scores of the first axis to represent the overall response variable, enabling further analysis with other factors [57]. For specific data, refer to Appendix A and Appendix A. Linear regression analysis was employed to investigate the impact of the seven different combinations of influencing factors on community diversity and phylogenetic diversity.

### 4.6. Data Analysis

To investigate the effects of islands on taxonomic and phylogenetic diversity as well as structure, we first standardized all taxonomic diversity indices (SR, Shannon–Wiener, Simpson, Margalef, and Pielou) and phylogenetic diversity (*SES.PD*) for all the Pingtan islands using the Z-score method. After confirming normal distribution through the Shapiro–Wilk test, we used Pearson correlation analysis to explore the relationships between island taxonomic diversity, phylogenetic diversity (*SES.PD*), and phylogenetic structure. Due to simplicity and effectiveness of Pearson’s correlation in measuring the linear relationship between normally distributed variables, which aligns well with our study objectives [58]. Additionally, we utilized a General Linear Model (GLM) to examine the island effects on plant community taxonomic diversity, phylogenetic diversity, and phylogenetic structure in Pingtan, because of the flexibility of GLM in handling different types of data distributions and its robustness in modeling the effects of multiple predictors on response variables [58]. Prior to the GLM analysis, we ensured that the data met the assumptions of regression analysis by checking for normality of data residuals, homoscedasticity of errors, and independence. In cases where island area and isolation effects were present, we observed a significant positive correlation between taxonomic, phylogenetic diversity, and structural indices as island area increased, while a significant negative correlation was observed with increasing island isolation.

Hierarchical partitioning analysis was used to assess the relative importance of predictor variables in explaining variation in a response variable. In hierarchical partitioning analysis, the total variation in the response variable is partitioned into unique and shared contributions from each predictor variable. The analysis proceeds in several steps, (1) Model Fitting: A regression model is fitted to the data, with the response variable regressed on all predictor variables of interest. (2) Partitioning of Variance: The total variance in the response variable is decomposed into components attributed to each predictor variable individually, as well as to combinations of predictor variables. (3) Calculation of Importance Metrics: Importance metrics are calculated for each predictor variable based on its unique contribution to the explained variance. (4) Interpretation: Finally, the results are interpreted to assess the relative importance of each predictor variable in explaining the variation in the response variable. In order to investigate the importance of island effects, island habitat heterogeneity, and anthropogenic disturbances on the taxonomic, phylogenetic diversity, and structure of plant communities on the Pingtan Islands, we selected combination factors that showed significance in influencing taxonomic and phylogenetic diversity indices. Through hierarchical partitioning analysis, we determined the relative contributions of significant island effects, island spatial characteristics, anthropogenic disturbances, and their combinations to taxonomic and phylogenetic diversity indices. Variables with a larger effect size (regression coefficients) and confidence intervals that do not include zero were considered to have a stronger impact and were used to identify factors with positive or negative effects [59].

All data analyses were carried out in R 4.0.5, using various packages for different purposes. The “*mass*” package was used for normality tests, “*corrplot*” for correlation analysis, “*picante*” for null model analysis, “*stats*” for linear analysis, “*hier.part*” for hierarchical partitioning analysis, and “*stats*” for PCA [51,58].

## 5. Conclusions

Island effects, especially island area effects, significantly influence both taxonomic and phylogenetic diversity on the Pingtan islands. Island effects and habitat heterogeneity contribute to increased diversity, while anthropogenic disturbances have the opposite effect. Additionally, human activities negatively impact island effects and habitat heterogeneity on diversity. However, our focus on taxonomic and phylogenetic diversity primarily considered island effects, habitat heterogeneity, and anthropogenic disturbances, without fully accounting for factors like soil characteristics and climate change. Soil characteristics directly influence plant growth and distribution, warranting comprehensive consideration in island biogeography studies. Exploring the interactions between invasive and native species, as well as the impacts of climate change on plant distribution and ecosystem dynamics, are crucial for understanding island diversity formation. Future research should delve into these factors to gain a more holistic understanding of island ecosystems.

## Figures and Tables

**Figure 1 plants-13-01537-f001:**
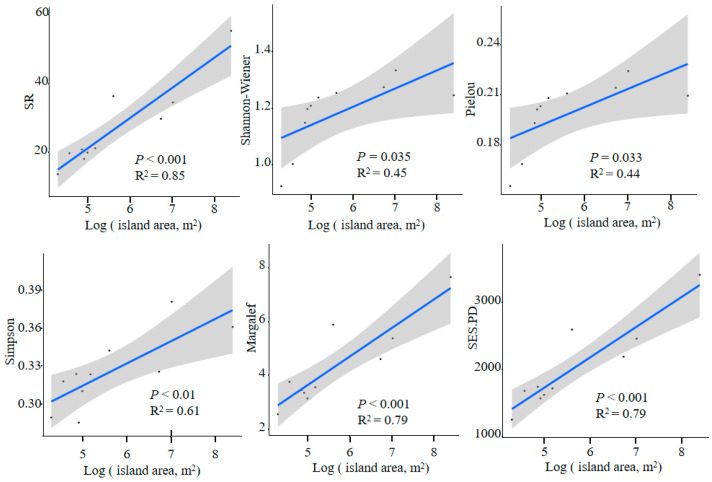
The linear relationship between island area and taxonomic and phylogenetic diversity. *SES.PD*, Standardized Effect Size of Phylogenetic Diversity. SR, species richness. The blue line represent the best-fit linear regression lines, and the gray area indicates the 95% confidence interval around the lines.

**Figure 2 plants-13-01537-f002:**
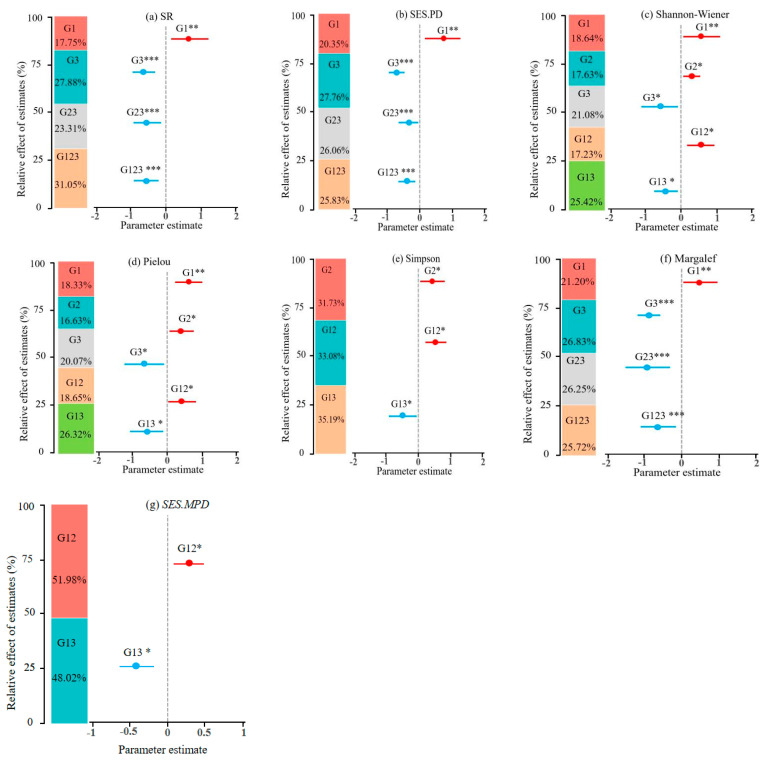
(**a**–**g**), Contributions of island effect, island habitat heterogeneity, and anthropogenic disturbances to taxonomic, phylogenetic diversity and structure. Lines represent 95% confidence intervals, red circles indicate a significant positive effect, blue circles indicate a significant negative effect. *SES.PD*, Standardized Effect Size of Phylogenetic Diversity. SR, species richness. *SES.MPD*, Standardized Effect Size of Mean Pairwise Distance. G1, island effect (island area + isolation). G2, island habitat heterogeneity. G3, human disturbance. G12, PCA values of island effect + island habitat heterogeneity. G13, PCA values of island effect + human disturbance. G23, PCA values of island habitat heterogeneity + human disturbance. G123, PCA values of island effect + island habitat heterogeneity + anthropogenic disturbances. * *p* < 0.05, ** *p* < 0.01, *** *p* < 0.001.

**Figure 3 plants-13-01537-f003:**
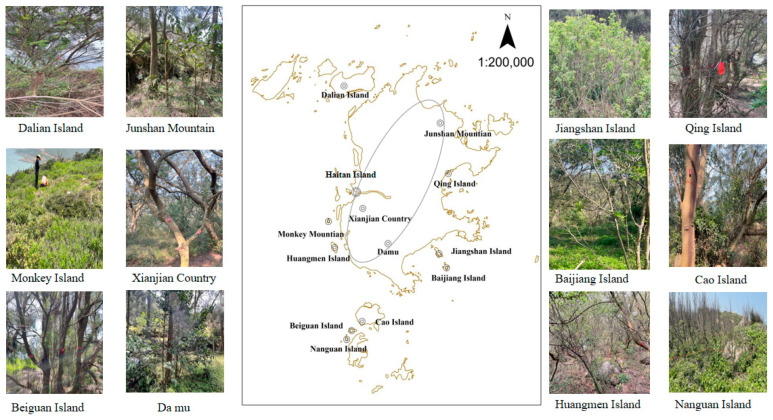
Overview of fixed plots on the Pingtan islands (Junshan Mountain, Xianjian Country, and Da mu are terrestrial areas of Haitan Island).

**Table 1 plants-13-01537-t001:** Linear analysis results of different combinations (island effects, island habitat heterogeneity, and anthropogenic disturbances) on taxonomic and phylogenetic diversity. SR, species richness. *SES.PD*, Standardized Effect Size of Phylogenetic Diversity. *SES.MPD*, Standardized Effect Size of Mean Pairwise Distance. G1, island effect (island area + isolation). G2, island habitat heterogeneity. G3, anthropogenic disturbances. G12, PCA values of island effect + island habitat heterogeneity. G13, PCA values of island effect + anthropogenic disturbances. G23, PCA values of island habitat heterogeneity + anthropogenic disturbances. G123, PCA values of island effect + island habitat heterogeneity + anthropogenic disturbances. * *p* < 0.05, ** *p* < 0.01, *** *p* < 0.001. Non-significant results are detailed in Appendix A.

Taxonomy Diversity Index	PCA-Variable	Estimate	R^2^	T-Value	*p*-Value
*SES.MPD*	G12	0.298	0.469	3.150	0.026 *
	G13	−0.351	0.522	−2.148	0.023 *
SR	G1	0.699	0.620	3.611	0.007 **
	G3	−0.819	0.948	−12.05	0.000 ***
	G23	−0.623	0.931	−10.38	0.000 ***
	G123	−0.471	0.925	−9.966	0.000 ***
*SES.PD*	G1	0.728	0.671	4.044	0.004 **
	G3	−0.815	0.939	−11.11	0.000 ***
	G23	−0.465	0.901	−8.542	0.000 ***
	G123	−0.414	0.896	−8.286	0.000 ***
Shannon–Wiener	G1	0.710	0.639	3.768	0.006 **
	G2	0.425	0.529	2.999	0.017 *
	G3	−0.589	0.489	−2.772	0.024 *
	G12	0.366	0.529	3.003	0.017 *
	G13	−0.477	0.545	−3.098	0.015 *
Pielou	G1	0.710	0.546	3.545	0.004 **
	G2	0.425	0.413	3.019	0.015 *
	G3	−0.589	0.351	−2.561	0.021 *
	G12	0.326	0.303	2.855	0.022 *
	G13	−0.517	0.438	−2.974	0.011 *
Simpson	G2	0.374	0.410	2.360	0.046 *
	G12	0.327	0.423	2.423	0.042 *
	G13	−0.325	0.432	−2.469	0.039 *
Margalef	G1	0.728	0.673	4.059	0.004 **
	G3	−0.855	0.914	−9.278	0.000 ***
	G23	−0.905	0.889	−8.006	0.000 ***
	G123	−0.655	0.880	−7.661	0.000 ***

## Data Availability

The data presented in this study are available on request from the corresponding author.

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
