# Peer review of "Anthropogenic Disturbances Influenced the Island Effect on Both Taxonomic and Phylogenetic Diversity on Subtropical Islands, Pingtan, China"

_plants, 2024, doi:10.3390/plants13111537_

Round 1
Reviewer 1 Report
Comments and Suggestions for Authors
In general, I like the manuscript. But I believe they have messed up with the different formulae they have employed. I will be more specific:
First of all, (Ni/N) = Pi
Second, Simpson's formula is incorrect: D = Sum[(Pi)2]
Third, Pielou 's index of evenness is J' = H'/Hmax and H' is the Shannon Diversity index. Hmax = lnS where S is Species Richness
All formulae should be corrected and calculations should be repeated and the new results should be discussed.
Author Response
Dear editor and reviewers,
We would like to express our sincere gratitude for taking the time to review our manuscript and for your insightful and constructive comments. Your thoughtful suggestions have greatly helped us to improve the quality of our manuscript. Based on these comments and suggestions, we have made careful modifications on our manuscript. The changes based on reviewers were marked in red color. We hope the revised manuscript will meet your standard. On behalf of all the authors of this article, I will answer questions and comments one by one. Once again, thank you for your time and effort in reviewing our manuscript again.
Best regards!
Sincerely yours,
Zhongsheng He and Jinfu Liu
College of Forestry
Fujian Agriculture and Forestry University
May 29, 2024
Comments and Suggestions for Authors
- But I believe they have messed up with the different formulae they have employed. I will be more specific: First of all, (Ni/N) = Pi. Second, Simpson's formula is incorrect: D = Sum[(Pi)2]. Third, Pielou 's index of evenness is J' = H'/Hmax and H' is the Shannon Diversity index. Hmax = lnS where S is Species Richness. All formulae should be corrected and calculations should be repeated and the new results should be discussed.
Reply: Thanks for your suggestions. We did indeed incorrectly write some of the diversity index formulas in our report. However, we used R language for the calculations, so the results were not affected. The correct formulas have been updated in the latest version, and the calculation results remain unchanged.The R code used for the calculations is as follows:
setwd("D:\\Rrun\\0000")
library(openxlsx)
library(reshape2)
library(vegan)
setwd("D:\\Rrun\\20201023")
herb.data <- read.xlsx("D:/Rrun/herbplots.xlsx")
library(reshape2)
herb.mat <- acast(herb.data,
formula = plot ~ species ,
value.var = "abundance",
fill = 0)
Shannon.Wiener <- diversity(herb.mat, index = "shannon")
Shannon.Wiener
Simpson <- diversity(herb.mat, index = "simpson")
Simpson
S <- specnumber(herb.mat)
J <- Shannon.Wiener/log(S).
For specific modifications, please refer to response1

Reviewer 2 Report
Comments and Suggestions for Authors
The MS "Anthropogenic disturbances influenced the island effect on both taxonomic and phylogenetic diversity on subtropical islands, Pingtan, China " provides an interesting case of study of factors influencing islands biodiversity, including islands characteristics and the importance of phylogenetic diversity in conservation.
The introduction presents some repetitive structures. It should have a more concise structure.
While the introduction covers a lot of ground, it could be strengthened by a clearer focus on the specific study's purpose.
Please, briefly mention how the research will contribute to existing knowledge on island biodiversity.
In addition, it would be useful to highlight the unique characteristics of islands and their impact on biodiversity and set the introduction to the ETIB concept differently with a briefly mention of its limitations
Please, improve the aims: state the specific research question or objective
The data analysis section presents a robust methodology for investigating the effects of islands on plant diversity. To better clarify this section, an explanation for choosing Pearson's correlation and GLM over potentially suitable alternatives could strengthen the methodology.
The text mentions "increasing island isolation." Specify how isolation is being measured (distance to the mainland, etc.).
Overall, the discussion section provides a good foundation for understanding the study's results and their implications. Some aspects can be improved:
The use of phrases like "observable trend" and "may reflect" could be replaced with more conclusive language if supported by the data.
The authors reference relevant studies to support their interpretations and highlight the novelty of their combined analysis of island effects, habitat heterogeneity, and anthropogenic disturbances. Please, consider adding these references:
10.1016/j.ecolind.2017.12.009
10.1111/j.1365-2664.2012.02193.x
10.1017/S0376892917000108
10.1007/BF00048036
Author Response
Dear editor and reviewers,
We would like to express our sincere gratitude for taking the time to review our manuscript and for your insightful and constructive comments. Your thoughtful suggestions have greatly helped us to improve the quality of our manuscript. Based on these comments and suggestions, we have made careful modifications on our manuscript. The changes based on reviewers were marked in red color. We hope the revised manuscript will meet your standard. On behalf of all the authors of this article, I will answer questions and comments one by one. Once again, thank you for your time and effort in reviewing our manuscript again.
Best regards!
Sincerely yours,
Zhongsheng He and Jinfu Liu
College of Forestry
Fujian Agriculture and Forestry University
May 30, 2024
Reviewer: 2
Comments and Suggestions for Authors
(1)The introduction presents some repetitive structures. It should have a more concise structure.
Reply: Thank you for suggestions. We have carefully reviewed and made the necessary revisions based on your suggestions to improve the conciseness and efficiency of the text. For specific details of the changes, please see the introduction section of the document.Thank you again for your valuable advice.
(2)While the introduction covers a lot of ground, it could be strengthened by a clearer focus on the specific study's purpose.
Reply: Thank you for your valuable suggestions on the introduction of our manuscript. We have revised the introduction based on your advice, and it now more clearly highlights the specific purpose and main objectives of our study. This revision has helped us articulate the importance and anticipated contributions of our research more clearly. For specific details of the changes, please see the introduction section of the document.
(3)Please, briefly mention how the research will contribute to existing knowledge on island biodiversity.
Reply: This research will contribute to existing knowledge on island biodiversity by providing a deeper understanding of the interplay between phylogenetic diversity and ecosystem function within island environments. By emphasizing the importance of evolutionary history and species characteristics in conservation strategies, the study enhances the understanding of how historical, geographical, and ecological variables shape biodiversity. This approach not only fills gaps in current biodiversity conservation practices but also aids in developing more targeted, effective conservation strategies that account for both species richness and evolutionary uniqueness, thereby ensuring the sustainability of island ecosystems.
(4)In addition, it would be useful to highlight the unique characteristics of islands and their impact on biodiversity and set the introduction to the ETIB concept differently with a briefly mention of its limitations
Reply: Thank you for your insightful suggestions. We have incorporated a section that highlights the unique characteristics of islands and their significant impacts on biodiversity. Additionally, we have restructured the introduction to the ETIB to provide a clearer overview, including a brief mention of its limitations. These modifications enrich the contextual framework of our study and enhance the reader’s understanding of the complexities involved in island biodiversity.
L50-69: Resource distribution within islands correlates with island size; larger islands possess more ecological niches and resources, supporting a higher diversity of species. Additionally, isolation reduces colonization rates, and greater isolation enhances the likelihood of species extinction[4]. Island spatial characteristics include distinct geographic, topographical, and ecological attributes that result in unique ecosystem structures[5]. The interplay between these characteristics and anthropogenic disturbances is significant; the unique geographic and ecological traits of islands increase their vulnerability to human impacts, which in turn shape island ecosystem patterns[6]. These unique geographical features make each island a distinctive sample for biodiversity research. Although the ETIB provides a basic framework for understanding the relationship between species richness and island size, as well as distance from the mainland, the theory has its limitations. It fails to fully account for the impacts of characteristic factors such as habitat heterogeneity on islands and human disturbances on diversity.
(5)Please, improve the aims: state the specific research question or objective
Reply: Thank you for suggestions. We have revised the aims of our study to state the specific research questions as per your suggestion.
L110-122: Both community taxonomic and phylogenetic diversity are not exclusively influenced by island effects, geographic isolation, habitat heterogeneity, and human activities may affect the island taxonomic and phylogenetic diversity. Based on the above description, our study addressed two main questions: (1) What pattern of taxonomic and phylogenetic diversity is observed with island area and isolation? (2) How do the island effect, island habitat heterogeneity, anthropogenic disturbances, and the combination of these factors influence taxonomic and phylogenetic diversity? We also hypothesize that taxonomic and phylogenetic diversity will increase with island area and decrease with isolation; Island effect, island habitat heterogeneity, anthropogenic disturbances and combined of these factors can influence taxonomic and phylogenetic diversity, anthropogenic disturbances could affect the island characteristics on both taxonomic and phylogenetic diversity.
(6)The data analysis section presents a robust methodology for investigating the effects of islands on plant diversity. To better clarify this section, an explanation for choosing Pearson's correlation and GLM over potentially suitable alternatives could strengthen the methodology.
Reply: Thank you for your valuable feedback. We appreciate your suggestion to provide an explanation for our choice of Pearson's correlation and Generalized Linear Models (GLM) over other potential alternatives. We have chosen Pearson's correlation due to its simplicity and effectiveness in measuring the linear relationship between normally distributed variables, which aligns well with our study objectives. We have also chosen GLM for its flexibility in handling different types of data distributions This approach has allowed us to comprehensively assess the complex interactions between island characteristics and plant diversity. We have revised the data analysis section to include this rationale, enhancing the robustness and transparency of our methodology. Thank you once again for your valuable guidance.
L520-532: To investigate the effects of islands on taxonomic and phylogenetic diversity as well as structure, we first standardized all taxonomic diversity indices (SR, Shannon-Wiener, Simpson, Margalef, and Pielou) and phylogenetic diversity (SES.PD) for all Pingtan islands using the Z-Score method. After confirming normal distribution through the Shapiro-Wilk test, we used Pearson correlation analysis to explore the relationships between island taxonomic diversity, phylogenetic diversity (SES.PD), and phylogenetic structure. Due to simplicity and effectiveness of Pearson's correlation in measuring the linear relationship between normally distributed variables, which aligns well with our study objectives[58]. Additionally, we utilized a General Linear Model (GLM) to examine the island effects on plant community taxonomic diversity, phylogenetic diversity, and phylogenetic structure in Pingtan, because flexibility of GLM in handling different types of data distributions and its robustness in modeling the effects of multiple predictors on response variables[58].
(7)The text mentions "increasing island isolation." Specify how isolation is being measured (distance to the mainland, etc.).
Reply: Thank you for suggestions. We have used Geographic Information System (GIS) software to accurately measure the characteristics of the islands, including island area, coastline length, distance to the mainland and the nearest island, as well as island elevation. Specific methods include using GIS spatial analysis tools to calculate area and length, point-to-point functions to measure distances, and integrating Digital Elevation Models (DEM) to obtain elevation data. We have supplemented these specific methods in the text.
L458-474: The characteristics of the surveyed islands were calculated by Geographic Information System (GIS) software. Parameters such as island area, coastline length, distance to the mainland, nearest island distance, island elevation, and Shape Index (SI) were used to depict the island features. Specifically, island area was determined by spatial analysis tools within GIS that compute the total area based on the island’s boundary coordinates. Coastline length was measured using GIS tools that trace the contour lines of the island perimeter, ensuring an accurate reflection of the island’s interface with water bodies. The distance to the mainland (MD, Distance to the mainland) and nearest island distance—measuring the shortest straight-line distance from a particular island to its nearest neighboring island—were calculated using point-to-point measurement functionalities in GIS. Island elevation was assessed by integrating GIS with terrain data, such as Digital Elevation Models (DEM), to extract the highest elevation point on the island. The isolation of the islands was assessed using the distance to the mainland. This metric, along with the nearest island distance, plays a crucial role in assessing island spatial characteristics as it provides information about the proximity of one island to another, which can influence ecological processes such as dispersal and species interactions [55].
Specific comments:
(1)The use of phrases like "observable trend" and "may reflect" could be replaced with more conclusive language if supported by the data.
Reply: Thank you for your thorough review and valuable suggestions on our manuscript. Regarding the use of more definitive language to replace phrases such as 'observable trend' and 'may reflect', we have carefully revised our manuscript following your guidance. Based on the support of the data, we have replaced these terms with more precise language to express our research findings, thus enhancing the accuracy and persuasiveness of the article.
L212-214: However, there was a tendency for a decrease in taxonomic and phylogenetic diversity, and phylogenetic structure (Fig. S2).
L269-270: This reflected that, under the influence of human activities, the joint action of these factors leads to a negative response in the ecosystem.
(2)The authors reference relevant studies to support their interpretations and highlight the novelty of their combined analysis of island effects, habitat heterogeneity, and anthropogenic disturbances. Please, consider adding these references:
10.1016/j.ecolind.2017.12.009
10.1111/j.1365-2664.2012.02193.x
10.1017/S0376892917000108
10.1007/BF00048036
Reply: Thank you very much for highlighting the importance of these additional references. We have carefully reviewed the studies you recommended and found three of them to be extremely helpful in supporting our interpretations and in emphasizing the novelty of our combined analysis of island effects, habitat heterogeneity, and anthropogenic disturbances. Based on the contributions of these documents, we have appropriately added these references to our article.
L611-613: Fois, M.; Cuena-Lombraña, A.; Fenu, G.; Cogoni, D.; Bacchetta, G. Does a correlation exist between environmental suitability models and plant population parameters? An experimental approach to measure the influence of disturbances and environmental changes[J]. Ecol. Indic, 2018, 86: 1-8.
L644-646: Carmona, C.P.; Azcárate, F.M.0.; de Bello, F.; Ollero, H.S.; Lepš, J.; Peco, B.; Taxonomical and functional diversity turnover in Mediterranean grasslands: interactions between grazing, habitat type and rainfall[J]. J. Appl. Ecol, 2012, 49(5): 1084-1093.
L661: Legendre, P.; Fortin, M. J. Spatial pattern and ecological analysis[J]. Vegetatio, 1989, 80: 107-138.
